# GFN-SR: Symbolic Regression with Generative Flow Networks

**Sida Li**
The University of Chicago
listar2000@uchicago.edu

**Ioana Marinescu**
Princeton University
ioanam@princeton.edu

**Sebastian Musslick**
University of Osnabrück, Brown University
sebastian.musslick@uos.de

## Abstract

Symbolic regression (SR) is an area of interpretable machine learning that aims to identify mathematical expressions, often composed of simple functions, that best fit in a given set of covariates $X$ and response $y$. In recent years, deep symbolic regression (DSR) has emerged as a popular method in the field by leveraging deep reinforcement learning to solve the complicated combinatorial search problem. In this work, we propose an alternative framework (GFN-SR) to approach SR with deep learning. We model the construction of an expression tree as traversing through a directed acyclic graph (DAG) so that GFlowNet can learn a stochastic policy to generate such trees sequentially. Enhanced with an adaptive reward baseline, our method is capable of generating a diverse set of best-fitting expressions. Notably, we observe that GFN-SR outperforms other SR algorithms in noisy data regimes, owing to its ability to learn a distribution of rewards over a space of candidate solutions.

## 1 Introduction

As a rising field that has garnered growing interest in the AI for Science community, symbolic regression (SR) aims to solve the following problem: given a dataset of covariates $X \in \mathbb{R}^{n \times d}$ and response $y \in \mathbb{R}^n$ (where $n$ is sample size and $d$ is the dimension of data), SR seeks to search for an interpretable expression $f : \mathbb{R}^d \to \mathbb{R}$ that best explains the dataset. Specifically, $f$ is required to be a closed-form symbolic expression that consists of symbols from a pre-defined library $L$ containing *operators* like $\exp(\cdot)$, $\div$, and variables like $x_i, 1 \leq i \leq d$. By learning both the structure and parameters (i.e. constants and variables) of a mathematical expression, SR thus explores a far richer search space and is more flexible than traditional regression tasks (e.g. linear regression) with fixed functional forms. On the other hand, the solution of SR is naturally more interpretable and succinct than black-box function approximations (e.g. non-linear neural networks), making it particularly useful in fields where interpretability matters, such as physics or cognitive science (Tenachi, Ibata, and Diakogiannis 2023; Musslick 2021).

Despite its appeal for automated scientific discovery, SR is widely acknowledged to be a challenging problem (Petersen et al. 2019; La Cava et al. 2021). In fact, due to the combinatorial nature of the underlying search problem, SR can be framed as an NP-hard problem (Virgolin and Pissis 2022). Historically, SR has been tackled through combinatorial optimization techniques like Genetic Programming (GP; Koza et al. 1989; Vladislavleva et al. 2008). GP maintains a population of symbolic expressions that evolves over time to enhance the best-fit function for a given dataset.

NeurIPS 2023 AI for Science Workshop.

However, GP is known to be sensitive to initialization and choice of hyper-parameters, and it scales poorly with the data size due to its computational complexity (Hassanat et al. 2018).

In recent years, there has been a flourishing of new SR methods. Although varying in many aspects, these methods coincidentally leverage the binary-tree representation of symbolic expressions. Bayesian Symbolic Regression (BSR) specifies the priors behind different operators and the likelihood of different modifications on the expression tree (Jin et al. 2020). It thus approximates the high-dimensional posterior distribution of suitable symbolic expressions for a dataset by taking Reversible Jump MCMC tree samples. On the other hand, AI-Feynman (Udrescu et al. 2020) relies on recursively exploiting the subtree of an expression and solving the simplified problems.

Most notably, Deep Symbolic Regression (DSR) exploits the discrete and sequential nature of building a valid expression tree (Petersen et al. 2019). Under a deep reinforcement learning framework, DSR models the construction process of an expression tree as querying a trajectory of "tokens" $\tau = [\tau_1, ..., \tau_T]$ (elements from the library $L$) one at a time from the categorical distribution $p(\tau_i | \theta, \tau_1, ..., \tau_{i-1})$, which derives from a neural network policy parametrized by $\theta$ (such as an RNN). The completed expression tree will then be evaluated based on its fit to the dataset (e.g. normalized RMSE), and the evaluation will be transformed into a reward function $R(\tau)$ (e.g. $1/(1 + NRMSE(\tau))$) to "reward" the trajectory. Ultimately, the objective is to optimize $\theta$ to maximize the objective: [1]

$$J(\theta) = \mathbb{E}_{\tau \sim p(\tau|\theta)}[R(\tau)] \tag{1}$$

i.e. the expected reward along the trajectory $\tau$, which translates to the expected fitness to the dataset for an expression generated by DSR.

Despite proving their effectiveness on SR benchmarks, DSR and its variants suffer from an inherent limitation within the RL setup: maximization of expected reward is often achieved through exploiting a single high-reward sequence of actions (Sutton et al. 1999). The issue becomes particularly critical under noisy settings, where equations with symbolic differences might produce datasets that appear deceptively similar. Therefore, the ability to generate diverse high-reward candidate equations— instead of clinging onto a single highest-reward solution—is a desirable property in such scenarios (E. Bengio et al. 2021).

In this paper, we present **GFN-SR**, a deep-learning SR method based on Generative Flow Networks (GFlowNets; GFN; Y. Bengio et al. 2021). GFN-SR sequentially samples tokens from a stochastic policy (represented by a deep neural net) to construct expression trees, similar to DSR. However, by casting SR into a GFlowNets problem instead of a return-maximization RL one, GFN-SR aims to align the distribution over the generated expressions proportional to their fit to the dataset, measured by a reward function $R(\cdot)$. Therefore, GFN-SR is able to not only identify a high-reward expression when one exists but also generate a set of diverse candidates when the reward distribution over the space of expressions is multi-modal.

To summarize our contributions: (1) we propose a novel deep-learning SR method by leveraging GFlowNets as sequential samplers, (2) we come up with an adaptive reward function that balances exploration with exploitation, (3) we report experiments evaluating our method's performance in settings with and without noise, showcasing its superior performance over other methods when the data are noisy.

## 2 Methods

### 2.1 SR under GFlowNets framework

In this section, we formalize the Symbolic Regression problem within the GFlowNets framework (Y. Bengio et al. 2021). We denote a **complete** expression tree as a binary tree representation of mathematical expressions with non-leaf nodes being operators or functions (in this paper, we use "operators" specifically for binary operators and "functions" for unary ones), and leaf nodes being variables and constants. Each token in the expression tree is part of a pre-defined library $L$. A complete expression tree $s$ can be reproduced from scratch: starting with an empty tree $s_0$, one can take the pre-order traversal of $s$ and sequentially add nodes to the tree. We can define $\mathcal{S}$ to be the

---

[1]In fact, the DSR paper adopted a risk-seeking version of this objective $J_{risk}(\theta) = \mathbb{E}_{\tau \sim p(\tau|\theta)}[R(\tau)|R(\tau) > R_\epsilon]$ ($\epsilon$ is a risk threshold) and demonstrated its superiority over vanilla policy gradient (VPG) that uses the objective above. Nevertheless, both objectives seek to maximize an expectation and suit the needs of this paper.

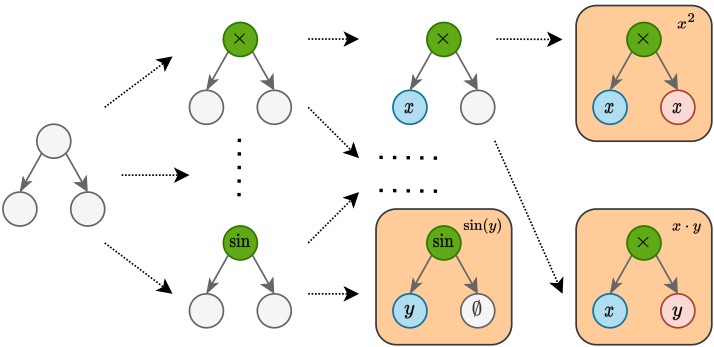

Figure 1: A part of the state space $\mathcal{S}$ and transitions between expression trees when we limit the maximum depth to two (see discussion of depth constraint below). The source state $s_0$ is an empty tree on the left, and terminal states are complete expression trees within orange plates. Any node in grey indicates that it has not been constructed yet. The token library $L$ includes, but is not limited to, $sin$, $\times$, and variables $x, y$.

overall set of complete expression trees and all intermediate trees in the above sequential construction process. Similarly, we denote $\mathcal{A}$ as the set of directed edges that represent all possible transitions between elements in $\mathcal{S}$ towards constructing some complete expression tree $s \in \mathcal{S}$. Trivially, since a transition can only happen when we add a new node to the previous intermediate tree, the combination $\{\mathcal{S}, \mathcal{A}\}$ creates a directed acyclic graph (DAG) for GFlowNets to be trained on (in fact, $\{\mathcal{S}, \mathcal{A}\}$ is a tree graph due to the pre-order traversal specification). In this DAG, $s_0$ is the source and any complete expression tree is a sink (terminal state). Figure 1 illustrates a small part of the state space $\mathcal{S}$ of expression trees and the possible transitions from an empty tree to a valid expression.

For the given DAG $\{\mathcal{S}, \mathcal{A}\}$ as well as a reward function $R(\cdot)$ acting on the terminal states in $\mathcal{S}$, GFN is a generative model with the objective to sequentially generate a complete expression tree $s \in \mathcal{S}$ with probability (E. Bengio et al. 2021)

$$\pi(s) \approx \frac{R(s)}{Z} = \frac{R(s)}{\sum_{s' \in \mathcal{S}^c} R(s')} \tag{2}$$

where $Z$ is a normalizing constant summing up all terminal rewards, and $\mathcal{S}^c \subset \mathcal{S}$ is the set of all complete expression trees. Due to the size of $\mathcal{S}^c$ being often intractable, GFN treats $Z$ as a learnable parameter. To achieve the objective in Equation 2, GFN also needs to learn a stochastic forward-sampling policy $P_F(\cdot|\cdot)$, which is represented by a deep neural network with parameter $\theta$. For a given state $s \in \mathcal{S}$, $P_F(\cdot|s)$ describes a categorical distribution with support $\{\tilde{s} \in \mathcal{S} : s \to \tilde{s} \in \mathcal{A}\}$. In our context, $P_F(\cdot|s)$ boils down to a categorical distribution over all possible tokens from $L$ and governs the next construction step from $s$.

To optimize parameters $\theta$ for the stochastic policy in GFN-SR, we adopt the *trajectory balance* (TB) loss proposed by Malkin et al. 2022. Let $\tau = (s_0 \to s_1 \to \cdots \to s_n = s)$ be the trajectory of generating a complete expression tree $s$ from scratch, we have:

$$\mathcal{L}_{TB}(\tau) = \left( \log \frac{Z_\theta \prod_{i=1}^n P_F(s_t|s_{t-1}; \theta)}{R(s) \prod_{i=1}^n P_B(s_{t-1}|s_t)} \right)^2 \tag{3}$$

where we condition both $Z$ and $P_F$ on $\theta$ to highlight their dependencies. The $P_B(s_{t-1}|s_t)$ term appearing in the denominator refers to a backward-sampling policy, which estimates the distribution of predecessors for a given state in the construction and is usually jointly learnt with $P_F$. Fortunately, the tree structure of $\{\mathcal{S}, \mathcal{A}\}$ in our problem setup gives each non-empty expression tree in $\mathcal{S}$ a unique predecessor (we can simply identify the most recently inserted node by reverting the pre-order and remove this node to obtain the predecessor). The TB loss for GFN-SR thus simplifies to:

$$\mathcal{L}_{TB}^*(\tau) = \left( \log \frac{Z_\theta \prod_{i=1}^n P_F(s_t|s_{t-1}; \theta)}{R(s)} \right)^2 \tag{4}$$

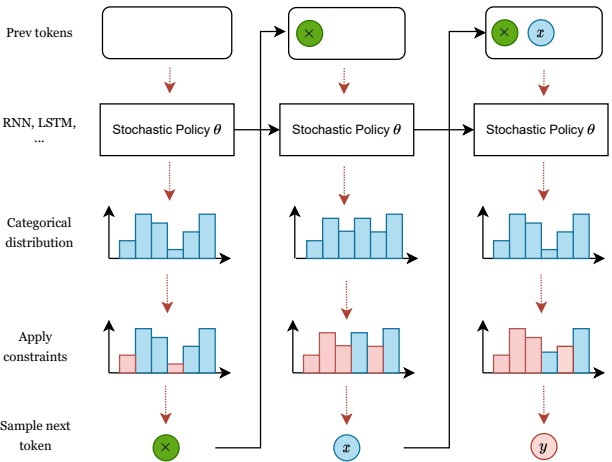

Figure 2: Illustration of the sequential expression tree construction. Through stochastic policy emits a raw categorical distribution, which is adjusted and re-normalized by the in-situ constraints (probabilities being zeroed out are labelled in red color). Finally, tokens for the next tree nodes are sampled and added to the growing expression tree following pre-order. This figure also corresponds to Figure 1, which views the same process but from a state space perspective.

Malkin et al. 2022 has shown that minimizing the TB loss is consistent with achieving the desirable objective in Equation 2. Finally, the stochastic policy is optimized via mini-batch gradient descent on $\theta$.

## 2.2 LSTM as the policy network

A key design choice in GFlowNets is the deep neural network behind the stochastic (forward) policy. As our goal is to sequentially predict the next token (symbol in our library $L$), we take inspiration from recurrent neural network (RNN) models that have been widely adopted by deep SR methods for similar sequence prediction tasks (Petersen et al. 2019, Landajuela et al. 2022). In particular, GFN-SR leverages Long Short-Term Memory (LSTM) networks (Hochreiter and Schmidhuber 1997). Through the inclusion of gating mechanisms (input, forget, and output gates) for the memory cell state, LSTMs can learn longer-term dependencies in the input sequences (Van Houdt, Mosquera, and Nápoles 2020). This facet of LSTMs aligns with our need to construct subsequent segments of expression trees based on prior, and potentially distant, tokens. In our implementation, we feed the one-hot representation of parent and sibling tokens of the next token to be generated into our LSTM policy network [2]. The LSTM then emits a probability vector of length equal to the size of library $L$ that represents a categorical distribution over the next token. Details on our LSTM architecture and hyper-parameters are listed in the Appendix.

## 2.3 Applying constraints to reduce computational complexity

Before a token is sampled out based on the probabilities emitted by the policy network, we apply *in-situ* constraints on the space of possible tokens by zeroing out certain entries in the probability vector and reweighing it. These constraints effectively reduce the computational complexity and accelerate the search for high-quality candidates. They are rules based on realistic assumptions that fall into the following categories:

**Composition constraints** We disallow certain compositions of operators and functions since they rarely exist in real equations (Petersen et al. 2019). For instance, a trigonometric function can not be nested in another one so expressions like $\sin(\cos(x) + \sin(x))$ is ruled out. We also prohibit functions that are inverse with each other to be composed directly (e.g. $(\sqrt{x})^2$).

---

[2]due to the pre-order, the parent token exists for any node but the root, and sibling exist for all nodes as the right child. We use a placeholder empty token if either does not exist.

**Depth constraints** We constrain how long an expression can be by imposing constraints on the depth (or height) of its corresponding expression tree. When the next node of a tree under construction is at maximum depth, we zero out all probabilities except for tokens representing variables and the constant symbol. This constraint is especially useful since we often process a batch of expressions together and the longest expression dictates the computation speed through the policy network.

**Constant constraints** Any function (i.e. unary operator) cannot take constant as its argument since that would simply represent another constant; any binary operator can only take constant as its right child, which (1) reduces expression redundancies due to the symmetry of tree structure and (2) prevents both arguments of such operator to be constants. The constant tokens are specially handled in GFN-SR as they are not assigned specific values during construction, and we instead optimize them in a separate process after full expressions are built (details in Appendix C). These constraints thus save a considerable amount of time fitting constants.

Figure 2 uses a concrete example to illustrate the two procedures (2.2, 2.3) above in sequentially generating an expression tree.

## 2.4 Reward baseline for GFlowNets

In this section, we discuss the role of reward function $R$ and a novel design of reward baseline for GFlowNets (the terminology "baseline" is inspired by Weaver and Tao 2013).

Based on Equation 2, a properly trained GFlowNet learns to generate a terminal state $s \in \mathcal{S}$ with probability $\pi(s) \propto R(s)$, thus the reward function should reflect our satisfaction on choosing $s$ as a solution to our problem. For the SR task, a reward function assigns a non-negative scalar to a terminal state (expression tree) $s \in \mathcal{S}$ to measure how well $s$ fits in the given dataset $(X, y)$. Usually, the reward function is some transformation on a loss function (e.g. RMSE, MAE) that quantifies such fitness. We assume $R(s) = 1/(1 + RMSE(s))$ as the "vanilla" reward function in the following discussion.

Generating a diverse set of high-quality candidate expressions requires a good balance between exploration and exploitation. GFlowNets are good at exploration by design since their training objective encourages generating expressions with probabilities proportional to the rewards. However, if the landscape of $R(s)$ is multi-modal yet the modes are not peaked enough, GFlowNets would find it difficult to concentrate on regions with high rewards. To address this issue, we can adjust $R$ such that low-reward expressions receive even lower rewards, while high-performing solutions are awarded greater rewards to emphasize their superiority. As a result, the distribution of $R$ would concentrate around the modes. Denote $B > 0$ as a baseline that does not depend on $s$, and consider:

$$R_B(s) = R(s)\left(1 + \gamma\left(\frac{R(s)}{B} - 1\right)\right) \tag{5}$$

where $\gamma \in [0, 1]$ is a scaling hyper-parameter that controls the degree of adjustment. We can easily verify that $R_B(s)$ is non-negative and satisfies the aforementioned properties. In the actual implementation, $B$ is initialized to be the average vanilla reward of expressions in the first batch. As training proceeds, we gradually increase $B$ toward the average vanilla reward for the top-performing expressions seen so far. The pseudo-codes for calculating the baseline and adjusted reward are included in Appendix A. Source code for GFN-SR is available at `https://github.com/listar2000/gfn_sr`.

## 3 Experiments

### 3.1 Noiseless Nguyen benchmark

We first experiment GFN-SR on the Nguyen benchmark problems (Uy et al. 2011) to evaluate its general capacity in fitting the given dataset (in terms of RMSE) when there is no noise. The Nguyen benchmark includes 12 distinct expressions of various complexity (see below) and has been widely

used to evaluate SR methods.

$$h_1 = x^3 + x^2 + x \qquad\qquad h_2 = x^4 + x^3 + x^2 + x$$
$$h_3 = x^5 + x^4 + x^3 + x^2 + x \qquad\qquad h_4 = x^6 + x^5 + x^4 + x^3 + x^2 + x$$
$$h_5 = \sin(x^2)\cos(x) - 1 \qquad\qquad h_6 = \sin(x) + \sin(x + x^2)$$
$$h_7 = \log(x+1) + \log(x^2+1) \qquad\qquad h_8 = \sqrt{x}$$
$$h_9 = \sin(x) + \sin(y^2) \qquad\qquad h_{10} = 2\sin(x)\cos(y)$$
$$h_{11} = x^y \qquad\qquad h_{12} = x^4 - x^3 + \frac{y^2}{2} - y$$

For comparison, we also run the same benchmarks on three other SR methods: GP, DSR, and BSR. Each of these methods has its unique hyper-parameters and architectures, and we tune them deliberately so that the total number of expression evaluations in one run (for one benchmark function) is around 25 million. We perform 20 repeating trials (but with different seeds) of the Nguyen benchmarks for each method and calculate the mean & standard error of the RMSEs (for detailed experimental setups for each method, see Appendix C).

Table 1: RMSE comparison of SR methods on Nguyen benchmarks
(standard error in parenthesis))

| Benchmark | Dataset | GFN-SR | DSR | GP | BSR |
|---|---|---|---|---|---|
| Nguyen-1 | U(20, -1, 1) | 0(0) | 0(0) | 0(0) | 0(0) |
| Nguyen-2 | U(20, -1, 1) | 0(0) | 0(0) | 0(0) | .002(.001) |
| Nguyen-3 | U(20, -1, 1) | 0(0) | 0(0) | 0(0) | .01(.004) |
| Nguyen-4 | U(20, -1, 1) | .02(.002) | **0(0)** | .05(.005) | .01(.003) |
| Nguyen-5 | U(20, -5, 5) | **.15(.021)** | .35(.028) | .51(.023) | .19(.066) |
| Nguyen-6 | U(20, -5, 5) | .25(.082) | **.012(.005)** | .90(.435) | .47(.158) |
| Nguyen-7 | U(20, 0, 2) | .128(.035) | .014(.0002) | .044(.002) | **6.2e-4 (2.9e-4)** |
| Nguyen-8 | U(20, 0, 4) | 0(0) | 0(0) | .091(0) | 9.6e-4 (4.6e-4) |
| Nguyen-9 | U(20, $[0,1]^2$) | .023(.015) | **0(0)** | .048(.011) | .001(.001) |
| Nguyen-10 | U(20, $[0,1]^2$) | .053(.014) | **.013(.019)** | .044(.006) | .025(.014) |
| Nguyen-11 | U(20, $[0,1]^2$) | .025(.012) | **.017(.021)** | .069(.012) | .054(.005) |
| Nguyen-12 | U(20, $[0,1]^2$) | **.037(.014)** | .040(.004) | .080(.021) | .082(.003) |

In general, results in 1 demonstrate the superiority of deep SR methods in terms of producing best-fitting solutions to a given dataset without noise. While DSR takes the lead for many benchmarks, we are thrilled to observe that GFN-SR, which is not trained under a return-maximization objective, still exhibits competitive performance and even outperforms other methods in Nguyen-5 and Nguyen-12. These results demonstrate the potential and efficacy of our new framework leveraging GFlowNets in symbolic regression tasks.

### 3.2 Noisy synthetic dataset

We set up a synthetic benchmark that is specially designed to evaluate an SR method's ability to recover exact symbolic expressions under noise. The benchmark consists of six equations that are grouped into three pairs (we denote equations in the $i$th pair as $f_i$ and $g_i$):

$$\text{Pair 1:} \quad f_1 = x^3 + x^2 + x \quad g_1 = \sin(x)(\sqrt{x} + \exp x) \quad x \in [0, 1]$$
$$\text{Pair 2:} \quad f_2 = \log(x + 1) \quad g_2 = 0.5 \cdot x \quad x \in [-0.1, 0.1]$$
$$\text{Pair 3:} \quad f_3 = x^2 - 4x + 3 \quad g_3 = 4 \cdot (1 - \sqrt{x}) \quad x \in [0, 1]$$

These equations are seemingly simple and solvable in noiseless settings (in fact $f_1$ is the first equation in the Nguyen benchmark). However, within the domains we specify, equations in each group are symbolically different yet very close to each other (see Appendix B for visualization). Therefore, if we further add noises to the dataset, it becomes challenging for an SR method to recover the right equation, especially if the method lacks the ability to explore a wide region of solutions and easily sticks around a single high-reward mode.

For each equation, the dataset consists of 40 points uniformly sampled from its specified domain (20 for training and 20 for testing) and responses $y$ under 10% noise level (i.e. noises $\epsilon \sim 0.1 \cdot N(0, Var(y))$). For each method (GFN-SR, DSR, and BSR), we repeat the experiment for 20 trials with different seeds and make sure that the number of equation evaluations in each trial is around 25 million for all methods. The recovery rate is defined as the percentage of trails (among all 20 runs) in which exact symbolic equivalence is achieved by the top predicted equations of a method (for DSR and BSR, this is achieved if any equation on the Pareto frontier matches the ground-truth; for GFN-SR, we compare the ground-truth with the most frequently occurring equations in the samples post-training, which is a stricter condition).

Table 2: Recovery rate of the noisy benchmark under 10% noise

| Equation | Group 1 | | Group 2 | | Group 3 | |
|---|---|---|---|---|---|---|
| | $f_1$ | $g_1$ | $f_2$ | $g_2$ | $f_3$ | $g_3$ |
| Dataset | $U(40, 0, 1)$ | | $U(40, -0.1, 0.1)$ | | $U(40, 0, 1)$ | |
| BSR | 30% | 5% | 0% | 90% | 5% | 75% |
| DSR | 75% | 5% | 5% | 40% | 5% | 90% |
| GFN-SR | **100%** | **15%** | **15%** | **95%** | **95%** | 85% |

Results in Table 2 show how SR methods are sensitive to the presence of noise, as the recovery rates are consistently low even for basic expressions like $f_2$. Bayesian Symbolic Regression (BSR) represents equations as a linear mixture of individual trees (Jin et al. 2020), thus it is not good at recovering equations exactly. DSR's return maximization objective, as mentioned earlier, sometimes misleads it to focus on the wrong but high-reward mode (i.e. the other expression in the group) while overlooking the correct one. Overall, GFN-SR is capable of generating a diverse set of candidate expressions and has outperformed the two other methods in recovering every equation except $g_3$.

## 4 Conclusion and future work

In this paper, we present GFN-SR—a novel symbolic regression method based on GFlowNets. The GFlowNets objective enables our method to explore and generate diverse candidate expressions, while the introduction of a reward baseline enables us to exploit high-reward regions only in the reward landscape. These exciting features are demonstrated through numerical experiments, where GFN-SR shows competitive performance in noiseless benchmarks and a superior ability to recover expressions under noise.

However, the proven success of GFN-SR is limited to experiments on the Nguyen benchmarks and our synthetic dataset. To further verify the effectiveness and understand the numerical properties of our approach, an immediate next step is to perform experiments on larger datasets, such as SRBench (La Cava et al. 2021) and Feynman SR database (Udrescu et al. 2020). We would also want to provide noisy data from real-world problems to confirm GFN-SR's exciting potential in recovering expressions from noise.

Another limitation of our current method is the enforcement of pre-order traversal in the tree construction process. Despite some aforementioned benefits (e.g. simplification of backward policy $P_B$), this restriction severely limits the number of possible transitions in the state space. If we allow growing any part of an incomplete tree, $\{\mathcal{S}, \mathcal{A}\}$ will still be a valid DAG that we can train GFlowNets on. This more flexible approach could potentially enhance the exploration of the expression space, though it may require additional computational time.

Finally, our presented method is versatile, and many of its components admit new ideas. For instance, Transformers have recently been the prevailing deep learning model for sequence generation tasks in NLP (Vaswani et al. 2017; Gillioz et al. 2020). They have the potential to replace RNNs as policy networks and provide a better understanding of global structures within symbolic expressions. Also, Landajuela et al. 2022 proposed a unified framework to combine deep-learning-based SR methods with other techniques, such as GP and polynomial regression. It is worth investigating to see how we can further improve GFN-SR under this framework.

## Acknowledgement

We thank Younes Strittmatter, Chad Williams, Ben Andrew, and other members of the Autonomous Empirical Research Group for their feedback and insights in this work. We appreciate the computation resources provided by the Center for Computation & Visualization (CCV) at Brown University. Sebastian Musslick was supported by Schmidt Science Fellows, in partnership with the Rhodes Trust, and supported by the Carney BRAINSTORM program at Brown University.

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

## Appendix A: Pseudocodes

---

**Algorithm 1** Calculate reward and update baseline

---

**Require:** a batch of expressions $\{s_1, \ldots, s_N\}$
**Require:** vanilla reward function $R$
**Require:** current baseline $B$
**Require:** a priority queue of best vanilla rewards $Q$

1: **Hyper-parameters:** scale parameter $\gamma$, moving average weight $\alpha$
$\qquad\qquad\qquad\qquad\qquad\qquad\qquad\qquad$ ▷ calculate the vanilla and baseline-aware reward
2: **for** $i$ in 1 to $N$ **do**
3: $\qquad R_i \leftarrow R(s_i)$
4: $\qquad R_i^* \leftarrow R(s_i) \times (1 + \gamma \times (\frac{R(s_i)}{B} - 1))$
5: **end for**
$\qquad\qquad\qquad\qquad\qquad\qquad$ ▷ update the priority queue with rewards sampled in this batch
6: $Q \leftarrow \text{updatePQ}(Q, \{R_1, \ldots, R_N\})$
$\qquad\qquad\qquad\qquad\qquad\qquad\qquad\qquad\qquad$ ▷ take the average of rewards in $Q$
7: $B' \leftarrow \text{calcAvgReward}(Q)$
8: $B^* \leftarrow \alpha \times B + (1 - \alpha) \times B'$

**Ensure:** rewards $\{R_1^*, \ldots, R_N^*\}$, new baseline $B^*$

---

## Appendix B: Dataset

Below is the plot of functions $f_1(x) = x^3 + x^2 + x$ and $g_1(x) = \sin(x)(\sqrt{x} + \exp(x))$ mentioned in Section 3.2, as well as the noisy observations for 20 sample points at noise level 10% (so they are similar to the training/testing dataset).

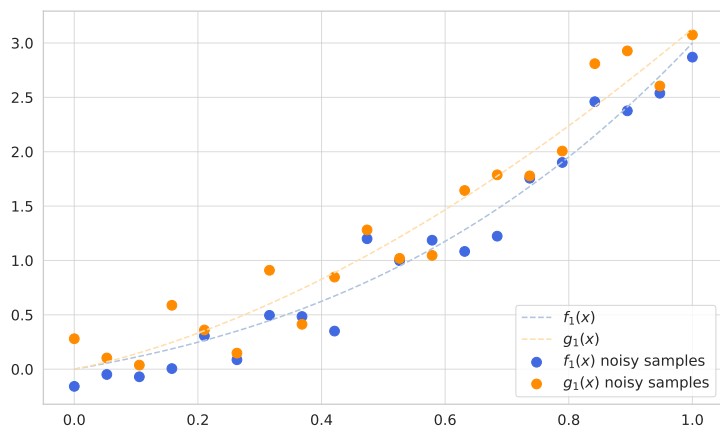

## Appendix C: Experiment Details

**Hyper-parameter tuning**  For GFN-SR, we consider three major hyper-parameters: (1) batch size $b \in [250, 500, 1000]$, learning rate $\alpha \in [\text{1e-4, 2e-4, 5e-4, 1e-3}]$, and the scaling parameter for reward baseline $\gamma \in [0.1, 0.2, \cdots, 0.9]$. We tuned them on a simplified Nguyen benchmark and the final hyper-parameters used are $b = 250, \alpha = \text{5e-4}, \gamma = 0.9$.

**LSTM architecture**  The LSTM policy network we use throughout the paper has 2 recurrent layers and 250 hidden states for each layer. We train the LSTM using Adam optimizer with the learning rate tuned above (Kingma and Ba 2014). Before we feed the parent and sibling tokens into the LSTM, they would go through a one-hot encoding.

**Token library**  Throughout all experiments for all methods, the library $L$ includes functions $\sqrt{\cdot}, \log, \exp, \sin, \cos, (\cdot)^2$, operators $\times, \div, +, -$, variables $x_1, \cdots, x_d$ for input dimension $d$, and the constant token. Certain SR methods require specifying the range of constants, which we specify as the maximum and minimum constant values in our benchmarks.

**Constant optimization**  In GFN-SR, a complete expression with at least one constant token cannot be evaluated by reward function $R$ directly. This is because these constants are not assigned any specific value during construction; instead, a constant optimization mechanism is applied. We have a dictionary and a counter. If an expression with constants has not been discovered before, or if its counter is at zero, we use BFGS (a numerical optimization algorithm) to find values of the constants that maximize $R$. We then cache its reward into the library and set its counter to some number that determines how frequently should we redo the optimization step. Otherwise, we simply read the cached reward for the expression from the dictionary and reduce the counter by one.

**DSR**  In our experiments, we use the deep symbolic optimization (DSO) Python package to run DSR method. Specifically, DSO improves the vanilla DSR method in Petersen et al. 2019 by providing certain post-processing. We use the same hyper-parameter configuration that Petersen et al. 2019 used in their paper.

**BSR**  Jin et al. 2020 have provided an official implementation of BSR; however, it is not continually maintained and certain results from the paper cannot be reproduced properly through this repository. We, therefore, use a third-party's implementation for the experiments. For BSR, we use a uniform prior for tokens in the library and a tree num $K = 3$.

**GP**  We use GPLearn, an open-source Python genetic programming toolkit, for bench-marking SR problems with GP. The specific parameters that I used are

```
SymbolicRegressor(population_size=5000, const_range=(-1, 2),
      generations=40, stopping_criteria=0.001,
      p_crossover=0.7, p_subtree_mutation=0.1,
      p_hoist_mutation=0.05, p_point_mutation=0.1,
      max_samples=1, verbose=1, init_method="full",
      parsimony_coefficient=0.01, random_state=0)
```

and the documentations is available here.

**Computation resource**  We run the above experiments using NVIDIA's Quadro RTX 6000 GPU and Intel's Xeon Gold 6242 CPU.

