# OpenReview forum: "GFN-SR: Symbolic Regression with Generative Flow Networks"
_NeurIPS.cc/2023/Workshop/AI4Science — NeurIPS2023-AI4Science Poster_

### Official Review · Reviewer_W7Xa · 2023-10-12
**Interesting work**

**Rating:** 6
**Confidence:** 2

**Review:**

The paper delves into enhancing symbolic regression (SR) with deep learning. Given SR's importance and complexity, traditional methods like Genetic Programming have shown limitations in scalability and sensitivity. DSR, using reinforcement learning, has recently been popular but can be restrictive by often converging to a single high-reward sequence, limiting diversity.

The introduced GFN-SR, which employs Generative Flow Networks, promises a more diverse set of high-reward solutions by matching the data distribution. Its strength lies in catering to noisy environments, offering multiple fitting expressions.

Preliminary experiments suggest GFN-SR's superiority, especially in noisy scenarios. It would be interesting to see broader validation.

---

### Official Review · Reviewer_qoMG · 2023-10-15
**Review comment for Submission156**

**Rating:** 6
**Confidence:** 4

**Review:**

In this paper, the authors propose GFN-SR, a deep learning-based symbolic regression (SR) method with Generative Flow Networks (GFN).
The proposed method is inspired by both Deep Symbolic Regression (DSR) and GFN. This paper also explicitly defines constraints applied to narrow down search space and reduce computational complexity, which is good, in deriving symbolic expressions to explain the given data.

The proposed method is assessed with 12 Nguyen datasets in terms of RMSE, and its overall performance seems comparable to DSR. While the authors acknowledge in Section 4, the reviewer believes that the evaluations in this study are very limited and should discuss numerical comparisons between the methods focused on symbolic regression's key properties, interpretability of the resulting solutions, besides "recovery rate" for six random datasets. e.g., Why didn't the authors use all the 12 Nguyen datasets?

Even though the reviewer understands that symbolic regression itself is tied well with AI for Science, this work should further discuss connections between this work and its application for science (e.g., in different research subjects such as physics) with some quantitative assessments.
On top of that, the Nguyen datasets use only 1-2 input variables with very small sample size and limited sampling range, thus the reviewer suggests more complex datasets suitable for AI4Science such as SRSD-Feynman datasets in "Rethinking Symbolic Regression Datasets and Benchmarks for Scientific Discovery" where DSR performed better than many existing methods.

P.S. What is "Due Each" ?

---

### Meta-Review · Area_Chair_wjzm · 2023-10-27

**Recommendation:** Accept (Poster)
**Confidence:** 2

**Metareview:**

The author(s) present(s) a deep symbolic regression (DSR) method based on Generative Flow Networks (GFN), which offers a more diverse set of candidates for deriving symbolic expressions from data. This approach is commendable for addressing the limitations of previous DSR methods that often converge to a single high-reward sequence of actions.

However, several concerns raised by reviewers remain:

- Broader validation is needed to verify the effectiveness of GFN-SR.
-  To align with the workshop's focus on AI for Science, it would be beneficial to include experiments or discussions related to the application of GFN-SR in scientific domains.